# Implementation of a cylindrical distribution function for the analysis of anisotropic molecular dynamics simulations

Richard J. Mandle[1,2]*

1 School of Physics and Astronomy, University of Leeds, Leeds, United Kingdom, 2 School of Chemistry, University of Leeds, Leeds, United Kingdom

* r.mandle@leeds.ac.uk

**Data Availability Statement:** All code and documentation for use are available from GitHub https://github.com/RichardMandle/cylindr.

**Funding:** a) UK Research and Innovation (UKRI) Future Leaders Fellowship to R. Mandle (grant no.

## Abstract

The cylindrical distribution function (CDF) is a convenient anisotropic analogue of the radial distribution function, the difference being the use of cylindrical shells for binning. As such, CDF analysis can be a powerful tool for the analysis of positional correlations within anisotropic systems, such as liquid crystals. Here we describe a lightweight Python tool, *cylindr*, for the calculation of cylindrical distribution function, which is compatible with the output of a number of popular MD engines. We demonstrate the use of *cylindr* in computing the CDF of a number of exemplar materials: classical and ferroelectric nematics; lamellar and columnar liquid crystals.

## Introduction

Molecular dynamics simulations are of ever-growing importance in liquid crystals and soft matter [1–4]; advances in hardware, software and force fields enable simulations of unprecedented scale and duration [5–9] that often closely approximate experimental data [10–13]. The radial distribution function (RDF, *g(r), Eq 1*) is frequently encountered in the analysis MD simulations of soft matter, and is the number of molecules within a spherical shell relative to that of a homogenous distribution

$$g(r) = \frac{V}{N4\pi r^2 \Delta r} \left\langle \sum_{ij}^{N} \delta(r - r_{ij}) \right\rangle \qquad \text{Eq 1}$$

Where $r$ is the distance between the selected atoms (or centres-of-mass)–$k$ and $m$–on the $i^{th}$ and $j^{th}$ molecules, respectively, $N$ is the number of molecules in the simulation, $r$ is the radius of the spherical shell, $\Delta r$ is the thickness of the spherical shell, and $\delta$ is the Dirac delta function. The angular brackets $(<...>)$ denote an ensemble average over the simulation. For anisotropic systems it is common for *g(r)* to be resolved into components parallel $(g_{\parallel}(r))$ and perpendicular $(g_{\perp}(r))$ to the director [14]

$$g_{\parallel}(r) = \frac{n(r)}{\rho_{bulk} V_{shell}} = \frac{n(r)}{\rho_{bulk} 2\pi r_w^2 dr_{\parallel}} \qquad \text{Eq 2}$$

MR/W006391/1); University of Leeds, University Academic Fellowship to R. Mandle (no grant number). b) The funders had no role in study design, data collection and analysis, decision to publish, or preparation of the manuscript. c) I receive my salary from the University of Leeds. d) We did receive funding for this work, see (a).

**Competing interests:** The authors have declared that no competing interests exist.

$$g_{\perp}(r) = \frac{n(r)}{\rho_{bulk}V_{shell}} = \frac{n(r)}{\rho_{bulk}2\pi dr_{\perp}^{2}}$$ 

Eq 3

Where $\rho_{bulk}$ is the mean density of the system and $n(r)$ is the number of molecules contained within a cylindrical volume (for $g_{\|}(r)$) of height $dr_{\|}$ and radius $r_{w}^{2} = (r_{cut}^{2} - r_{\|}^{2})^{1/2}$), or a cylindrical shell (for $g_{\perp}(r)$) with a height of $2(r_{cut}^{2} - r_{\perp}^{2})^{1/2}$ and width of $dr_{\perp}$, where $r_{cut}$ is the cut-off value used for calculating the distribution function.

The orientational-radial distribution functions $g_{1}(r)$ and $g_{2}(r)$ are sometimes encountered when considering anisotropic systems [15, 16]

$$g_{1}(r) = \langle \delta(r - r_{ij})(u_{i}.u_{j}) \rangle$$

Eq 4

$$g_{2}(r) = \left\langle \delta\left(r - r_{ij}\right)\left(\frac{3}{2}(u_{i}.u_{j})^{2} - \frac{1}{2}\right) \right\rangle$$

Eq 5

Where $u_{i}$ and $u_{j}$ are two vectors of unit length describing the orientation of the $i^{th}$ and $j^{th}$ molecules, respectively, and $r$ is the distance between the centre-of-mass (or selected atoms) of these same two molecules, with Eq 4 and Eq 5 corresponding to the Legendre polynomials.

The cylindrical distribution function (CDF) eschews the radial binning used for $g(r)$ and $g_{1}(r)$ in favour of cylindrical bins in order to preserve information on anisotropic positional order. The CDF is a logical progression from the anisotropic forms of the RDF shown in Eqs 2 and 3, as both parallel and perpendicular components are considered simultaneously, and so information about the spatial relationship between features is preserved. In its simplest form, the cylindrical distribution function is defined here as

$$g_{(h,r)} = \frac{\langle n(h,r) \rangle}{\rho_{bulk}V_{shell}}$$

Eq 6

Where $\langle n(h,r) \rangle$ is the density of particles in a cylindrical shell of height $h$ and inner radius $r$, normalised against the volume of the cylindrical shell, and the bulk density of particles in the system (Fig 1). There is interest in experimental measurement of the CDF for oriented systems [17–19], as well as its calculation from MD trajectories [20–22]. Here we describe an open source Python software tool (*cylindr*) for the calculation of CDFs from MD trajectories, and demonstrate its use in analysing several MD simulations of liquid crystalline materials with varying degrees of anisotropy.

## Implementation

Our CDF analysis code (*cylindr*) performs a number of sequential operations, outlined in Fig 1B. We elected to use MDtraj to read trajectory and topology information owing to its speed and also compatibility with the native output of many of the most popular MD engines [23].

*Cylindr* calculates cylindrical distribution functions (CDF) for a reference set of positions, either the centre of mass (-mode default), specified atom name or element name (-sel name, -sel element, respectively). By setting–mode hybrid, the CDF is computed between the centre-of-mass of the each and all neighbouring selections chosen with–sel, as above.

As a cylinder is anisotropic, the cylindrical binning procedure yields different results depending on the orientation of the cylinder length relative to the simulation. The default mode of *cylindr* is to compute the nematic director and orient the cylinder length along this vector, this can be oriented perpendicular to the director with–ori perp1 or–ori perp2.

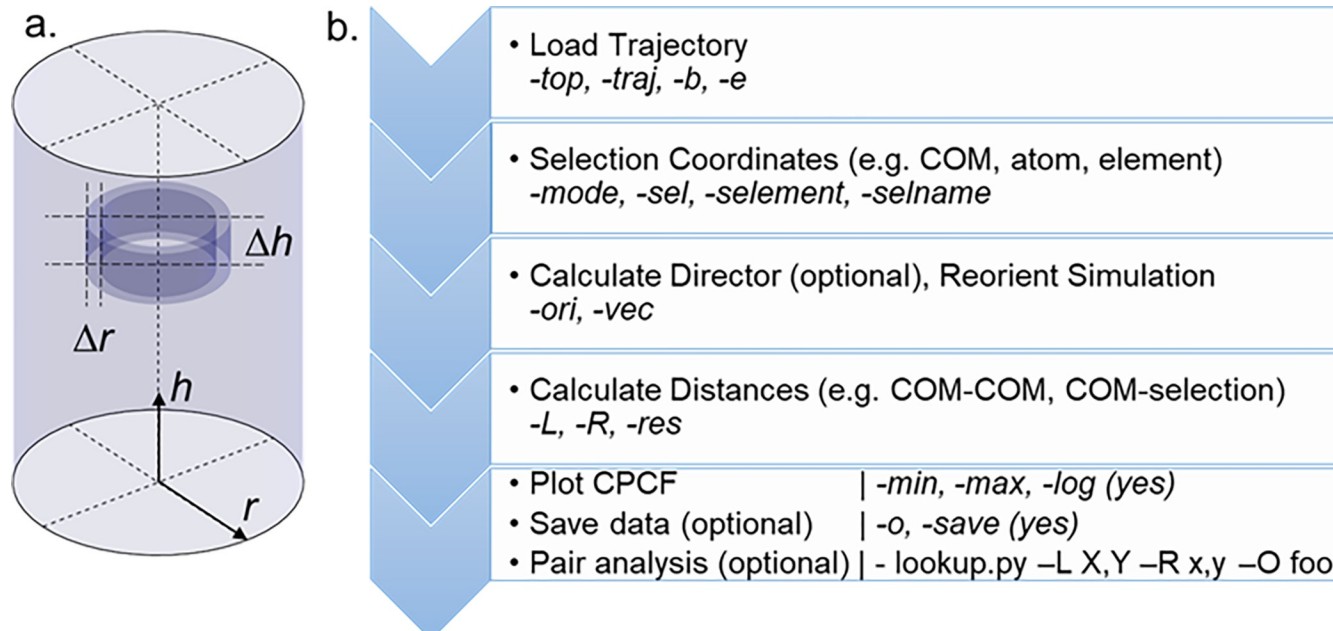

**Fig 1.** (a) Illustration of the cylindrical binning procedure, where Δr and Δh are the radial and length windows used in a given cylindrical bin; (b) Operations performed by *cylindr*, with user inputs shown in italics.

Additionally, the user can supply any 3-vector (with the flags–ori user and–vec *x,y,z*) for cylinder orientation.

Next, *cylindr* counts the number of centres-of-mass in a series of cylindrical shells for each frame, as well as the volume of each shell. We sum the CDF for each frame before normalizing against the shell volume and the average density. The volume of the cylindrical shells can be controlled using the user supplied length (-L) and radius (-R) limits and resolution (-res) inputs. Increasing either limit performs the CDF over a larger at the cost of increased compute time, and vice versa. The *resolution* variable determines the size of the cylindrical shells used for binning and is number of shell increments per angstrom—the default value of 4 was found to be sufficient for all instances in this work, a lower value gives coarser outputs at reduced computational cost, and vice versa.

The user can specify the minimum (-min) and maximum (-max) values used in the colourmap of the resulting CDF plot, with default values being zero and the maximum value, respectively. The CDF can be presented as log-normalized by using the–log flag.

The optional–save flag records all intermolecular distances in each frame as a compressed. npz file; the companion tool *lookup.py* takes user specified length (-L) and radius (-R) windows and returns the indices and time step of molecular pairs that satisfy these distance constraints, exporting as a.csv file. This enables quick visualisation of pairing modes that generate hits in the CDF plot using other software tools (VMD, nglviewer etc.).

## Simulation methodology

We present a number of exemplar simulations on which we perform CDF analysis. All simulations were performed in Gromacs 2019.2 [8, 24–29] with GPU acceleration via CUDA 10.1.168. Materials were chosen as representative examples of mesophases of interest: 5CB (nematic); RM554 (ferroelectric nematic); C5-Ph-ODBP-Ph-OC12 (bent core nematic); CB7CB (twist-bend nematic); 8OCB (smectic A); HAT-6 (hexagonal columnar). MD

simulations employed the GAFF force field with modifications for liquid crystalline molecules (GAFF-LCFF) [10], where available. For simulations of nematic and smectic liquid crystals we calculate the second-rank orientational order parameter <P2> *via* the Q-tensor approach [30] according to Eq 7

$$Q_{\alpha\beta} = \frac{1}{N} \sum_{m=1}^{N} \frac{3a_{m\alpha}a_{m\beta} - \delta_{\alpha\beta}}{2}$$                 Eq 7

where N is the number of monomers, m is the monomer number within a given simulation, $\alpha$ and $\beta$ represent the Cartesian x, y and z axes, delta is the Kronecker delta, *a* is a vector that describes the molecular long axis, which is computed for each monomer as the eigenvector associated with the smallest eigenvalue of the inertia tensor. The director at each frame was defined as the eigenvector associated with the largest eigenvalue of the ordering tensor. The order parameter <P2> corresponds to the largest eigenvalue of $Q_{\alpha\beta}$. For polar/ferroelectric nematics we introduce the polar order parameter, <P1>, which is calculated according to Eq 8

$$< P1 >= \langle cos\theta \rangle$$                 Eq 8

Where $\theta$ is the angle between the inertia tensor of a given molecule and the nematic director at a given frame. We compute the ferroelectric polarisation (Ps) of polar nematic simulations as the simulation dipole moment over the simulation volume. Quoted errors for order parameters and polarisations are one standard deviation from the mean.

## Results

4-Cyano-4'-pentylbiphenyl (5CB, Fig 2A) is perhaps the most well-studied nematic liquid crystal, both experimentally and *in silico*, and so is a logical demonstrator for our CDF code. We performed a 50 ns long simulation of 3600 molecules of 5CB at a temperature of 293 K, giving a nematic phase with an orientational order parameter (<P2>) of 0.56. We computed the CD, with the cylinder length both parallel (Fig 2D) and perpendicular (Fig 2E) to the director, as well as the RDF (Fig 2F) and its resolved variant (Fig 2G) for the production MD run.

Both CDF orientations show a region of zero probability centred at h = r = 0 Å, which results from steric repulsion of molecules. High probability regions at h = ±13, r = 0 Å and h = ±18, r = 0 Å result from antiparallel pairs of molecules, and at h = 0, r ~ 5.5 Å from side-by-side pairs. A series of arcs centred on h = 0 Å occur every r ~5.5 Å as a consequence of side-by-side packing of molecules; the first arc being the neighbouring molecule(s), the second being the neighbour of neighbours and so on. The diffusivity of all features indicates a lack of positional order, as expected for the nematic state. For the CDF oriented perpendicular to the director we see diffuse concentric rings every r ~5.5 Å, the origins of these being neighbouring molecules as discussed above.

Contrast this with the conventional RDF plot, shown in Fig 2F. Each of the features present in the CDF can be located in the RDF; the region of zero probability at r < 3.5 Å, the large peak at r ~ 5.5 Å, and sequential shells occurring every 5.5 Å. However, owing to the radial binning procedure information about anisotropy and orientation is lost. In the present case, calculation of the RDF took around 5 minutes using the Gromacs tool *gmx rdf*, whereas the CDF was computed in around 7 minutes using *cylinder.py*, both on a single Intel E5-2650v4 CPU core on the ARC3 computer at the University of Leeds.

By resolving the RDF into components perpendicular and parallel to the director (Fig 2F) we can see the same features present in the CDF. However, resolving the RDF in this way neglects off-axis interactions; a rather subtle example is the overlap of the 'on-axis' arcs (at h~

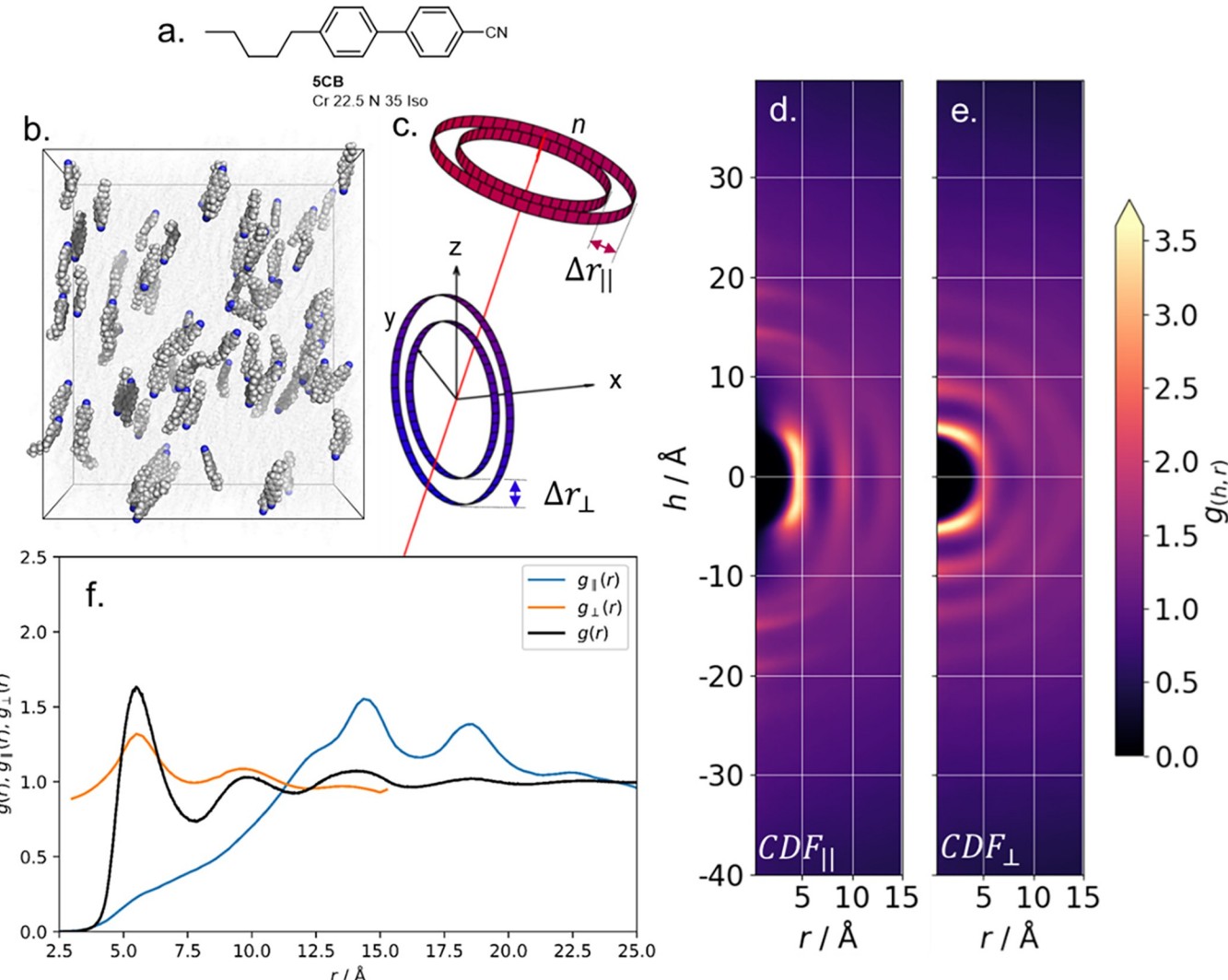

**Fig 2.** (a) Molecular structure and transition temperatures (°C) of 5CB; (b) instantaneous configuration of 5CB in the nematic phase at 293 K, the simulation has $<P2> = 0.56 \pm 0.019$ over the 50 ns MD run; (c) schematic showing the nematic director from (b) as a red arrow, and (exaggerated) cylindrical shells whose radius ($\Delta r$) forms a plane orthogonal ($\Delta r_{\parallel}$) or parallel ($\Delta r_{\perp}$) to the director, used to calculate the CDF parallel and perpendicular to the director, respectively; (d) parallel and (e) perpendicular CDF plots for the nematic simulation of 5CB as generated by *cylindr*; (f) radial distribution functions (RDF) for the same simulation, with the RDF resolved into parallel and perpendicular components according to Eq 2 and Eq 3.

10–12, r ~ 0–6 Å) with the off-axis arc that results from the lateral 'neighbour-of-neighbours' (at h ~ -10-10, r ~ 6–9 Å). This subtle anisotropy in local positioning manifests in variations in intensity in the concentric rings in the perpendicular CDF plot (Fig 2E) which is not apparent from the resolved RDF (Fig 2F).

So-called ferroelectric nematics ($N_F$) are an exciting recent discovery [31–33], in which the orientational order of conventional nematics (such as 5CB) is augmented with polar order. [34] Polar nematic order gives rise to extremely large ferroelectric polarisations [32], which are of interest to both fundamental and applied science [35–38]. We simulated a polar nematic configuration of the $N_F$ material RM554 (Fig 3A) [39] by taking an isotropic configuration of 680 molecules with liquid-like density of 1.1 g cm$^3$ and applying a static electric field (0.1 V nm$^{-1}$) along the y-axis; this gives a large polar order parameter ($<P1>$) of ~ 0.9 and an orientational order parameter ($<P2>$) of ~ 0.65. Following this, a production MD simulation

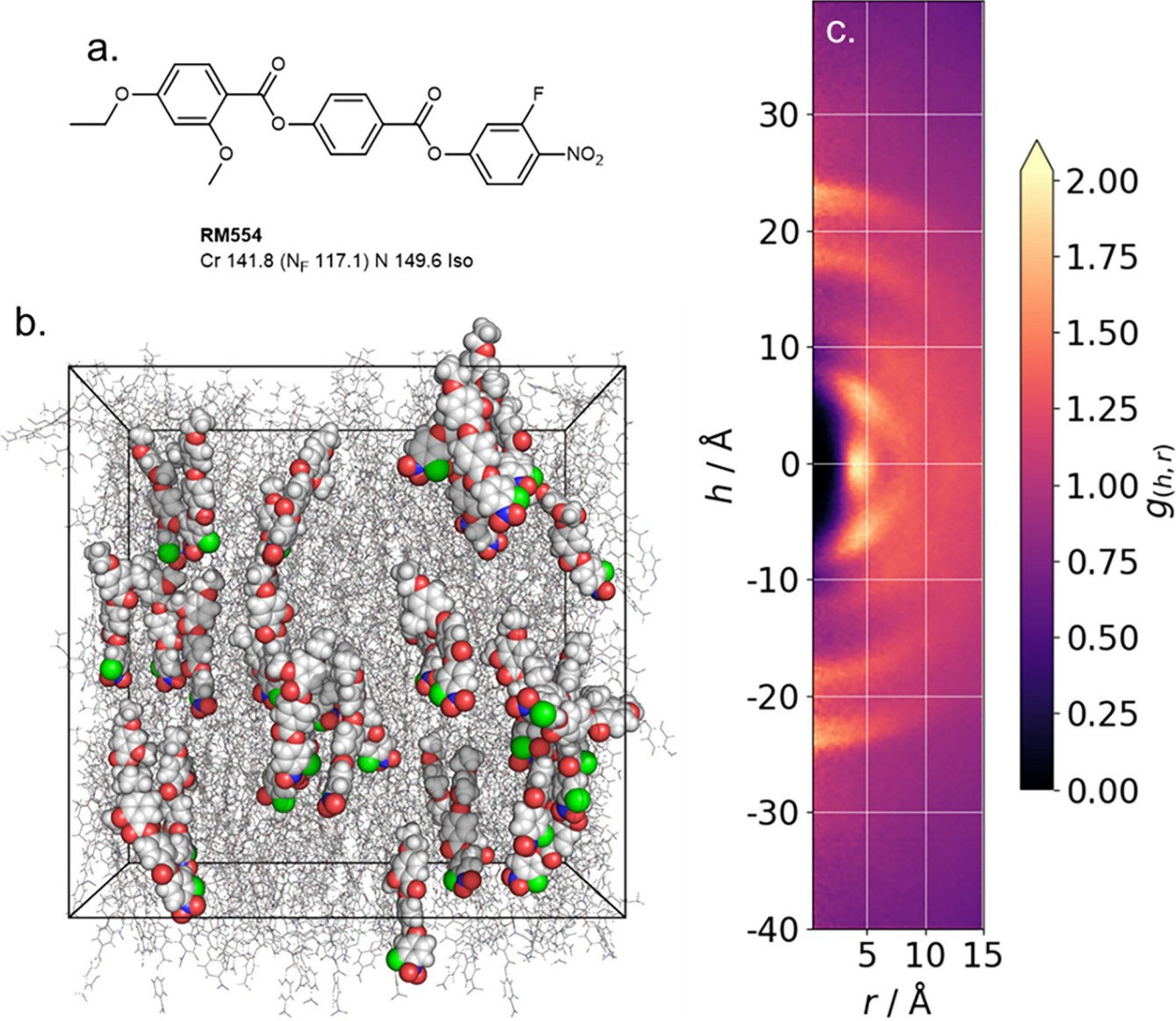

**Fig 3.** (a) Molecular structure and transition temperatures (˚C) of RM554 [39]; (b) instantaneous configuration of RM554 in the polar nematic configuration at 375 K, the simulation has the following average properties over the 250 ns MD run: $<P1> = 0.92\pm0.014$ $<P2> = 0.66\pm0.014$, Ps = 6.8±0.03 µC cm$^2$; (c) CDF for the polar nematic configuration of RM554 as generated by *cylinder*.

(without biasing field) of 250 ns was performed from this polar configuration at a temperature of 375 K with anisotropic pressure coupling; note that, as with prior simulations, the polar order is essentially constant over the entire production run [40].

Comparison of the CDF of the polar nematic phase of RM554 with the conventional nematic phase of 5CB reveals significant differences. Polar nematic phases predictably feature significant head-to-tail intermolecular associations, leading to arc-like features centred on r = 0 and h ~ ± 20 Å (i.e. the molecular length). Off-axis features correspond to staggered (h ~ ± 11 Å and ± 6 Å) and side-by-side (h = 0 Å) pairing modes.

We now consider nematic phases formed by bent core molecules [41], specifically the oxa-diazole C5-Ph-ODBP-Ph-OC12 (Fig 4A) [42–44]. Our simulation consists of 1000 molecules

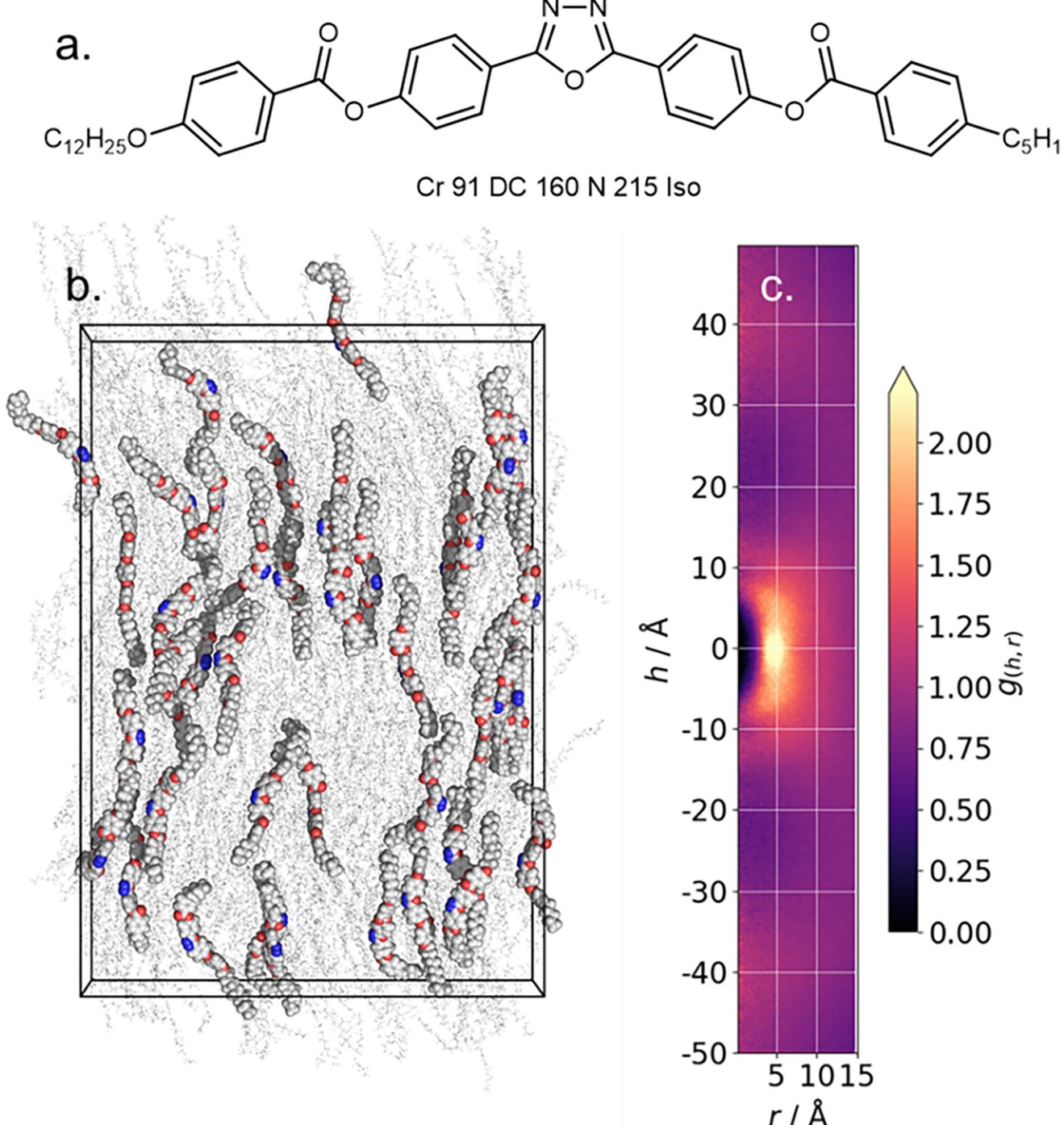

**Fig 4.** (a) Molecular structure and transition temperatures (˚C) of C5-Ph-ODBP-Ph-OC12 [42]; (b) Instantaneous configuration of the nematic phase of the same material (T = 463 K, t = 236 ns), (c) CDF plot produced over the entire production trajectory; the cylinder length cut-off was extended to 50 Å using the–L 50 flag.

at 483 K; a non-polar nematic phase forms spontaneously giving a $<P2>$ of 0.61±0.019, which compares favourably with experimental data and earlier MD simulations [45]. We compute the CDF, using a slightly larger length cut-off of 50 Å to account for the end-to-end molecular

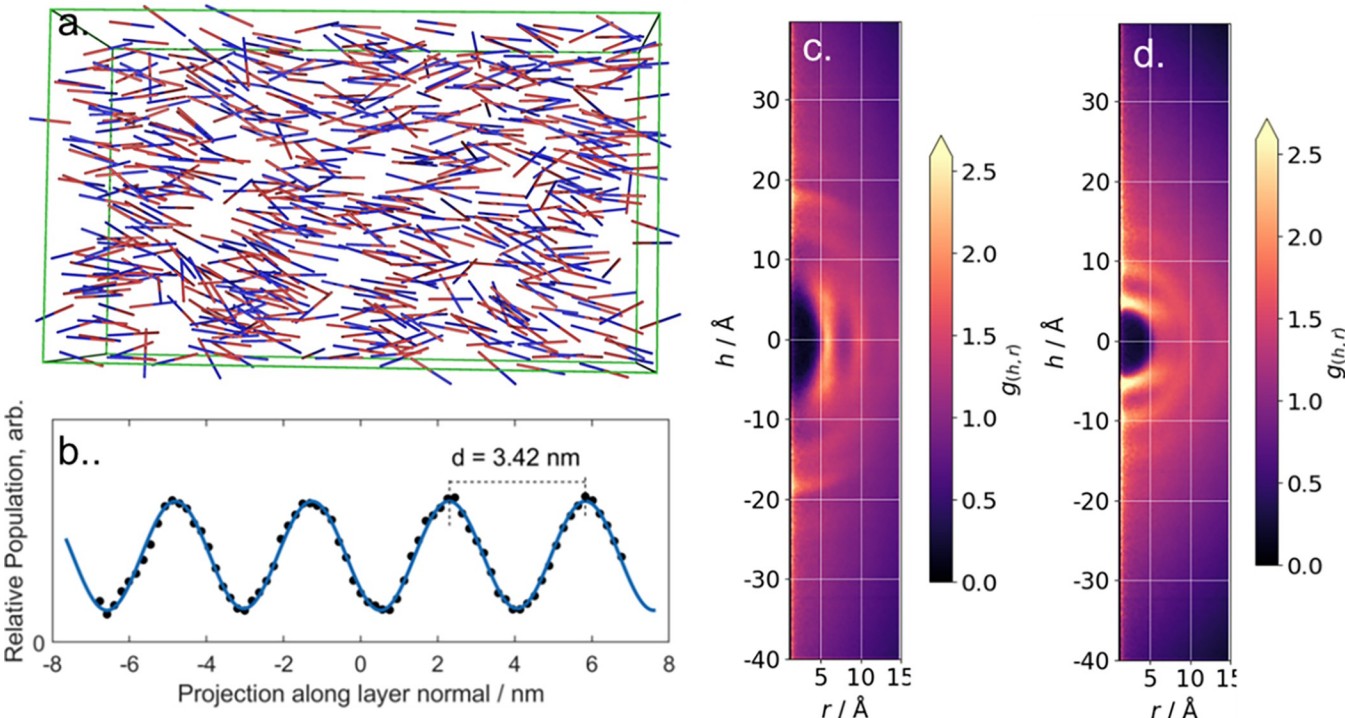

**Fig 5.** (a) Instantaneous configuration of the SmA mesophase formed by 8OCB (T = 365 K, t = 216 ns); here, to aid visualisation, we depict each molecule as a rod that extends from the nitrogen to the oxygen atom of 8OCB; (B) Calculated relative centre-of-mass populations along the layer normal using *gmx density* as described in the text; the solid line is a Fourier fit, presented as a guide to the eye. CDF plots generated with *Cylindr* for the SmA phase of 8OCB; the cylinder length being oriented along the layer normal (c) and in the layer plane (d), in both cases the orientation of the cylinder length was set with the–ori flag, specifying "nem" and "perp1" for orientation along and perpendicular to the layer normal, respectively.

length of this material (~ 45 Å). The most prominent feature in the CDF is at $h = 0$, $r = 5$ Å, and results from side-by-side pairs of molecules which are themselves likely to be consequence by dipole-dipole interactions between neighbouring 1,3,4-oxadiazole units; the lack of repeats indicates these interactions to be short-range in nature. A pair of diffuse features at $h \sim \pm 40$ Å result from the end-to-end separation of molecules, while diffuse features at $h \sim \pm 9$ Å are a consequence of staggered pairs of molecules with the 1,3,4-oxadiazole being proximal to the carboxylate ester.

We now turn to analysis of simulations of mesophases with some degree of positional order (layers, columns etc.). We simulated 1500 molecules of 8OCB at a temperature of 365 K; visual inspection reveals the formation of a smectic A mesophase after around 180 ns (Fig 5A); the production MD run was continued for a further 250 ns from this point, and only the smectic portion of the trajectory was utilised in our subsequent analysis. For the smectic portion of the trajectory we find $<P2> = 0.66 \pm 0.022$, close to both experimental [16, 46] and prior MD simulations. We confirmed the formation of a smectic A phase by dividing the simulation into 75 slices of equal thickness (with the slice plane being perpendicular to the director) and computing the number of centres-of-mass (COM) in each slice; a plot of COM density vs thickness gives the layer spacing as the distance between the maxima (Fig 5B). For our simulation we find an average layer spacing of 3.42 nm, which compares with an experimental value 3.2.

We first consider the CDF when the cylinder length is oriented along the layer normal (Fig 5C). Diffuse arcs at $h \sim 14$ and 18.5 Å are the smectic layer periodicity; recall, that n(O)CB materials form antiparallel pairs, thus, the centre-of-mass of a given molecule is not expected

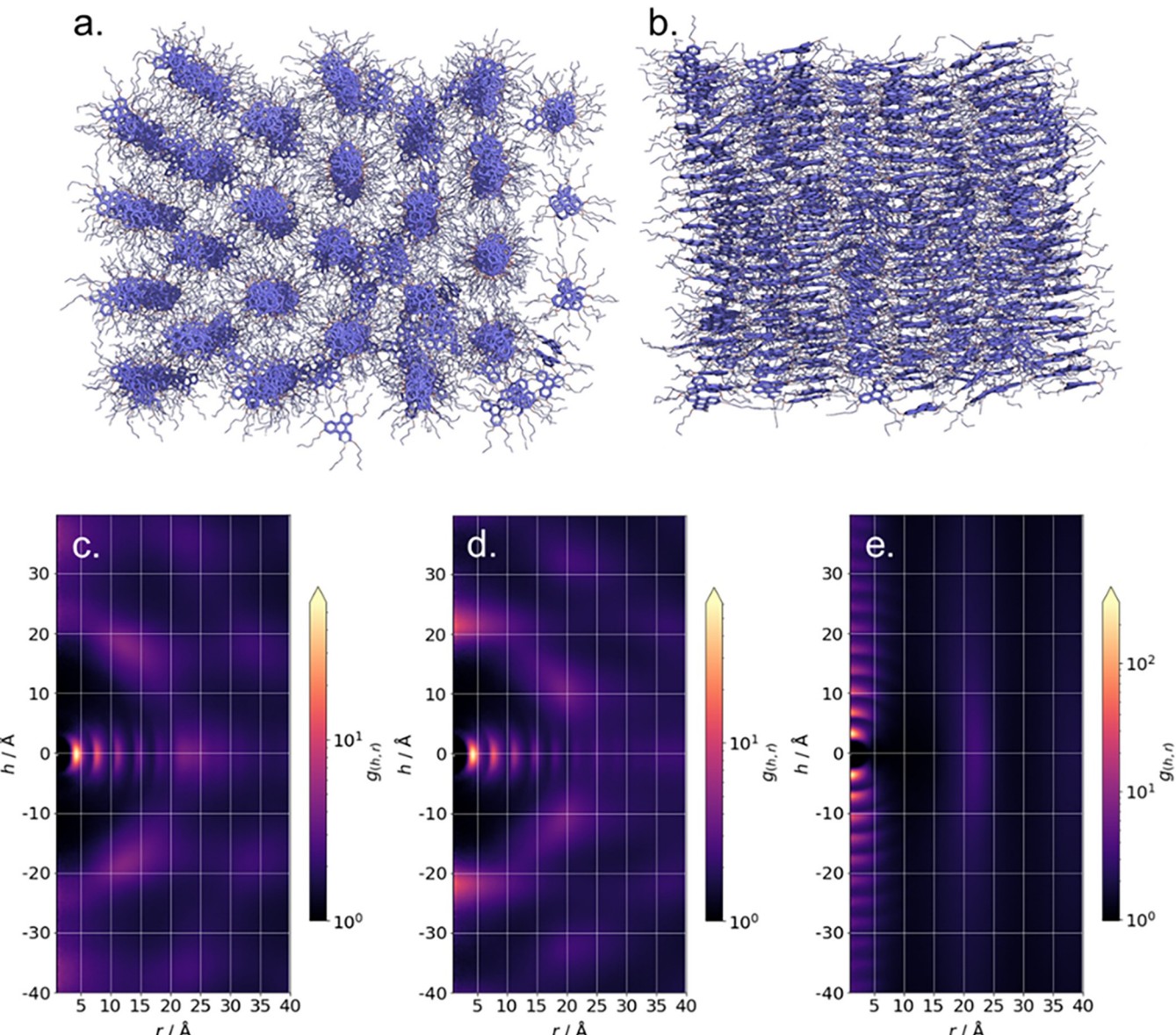

**Fig 6.** Instantaneous configuration of the hexagonal columnar phase formed by HAT6 at 68 ns viewed in the -xy plane (a) and -xz plane (b). Corresponding CDF plots generated using *cylindr* with the cylinder length oriented along the x- (c), y- (d) and z- (e) axes. Plots were generated using the following options– save no, -log yes, -ori user, -vec 1,0,0 (for c, 0,1,0 for (d), 0,0,1 for (e)).

to be equidistant from the two neighbouring layer interfaces. The diffusivity of the arcs that result from the layer periodicity consequence of the diffuse nature of the layers. Features centred at h = 0 Å and r = 6, 10, 14 (. . .) Å result from neighbouring molecules within the layers; the features grow more diffuse and less pronounced with increasing shell radius, pointing towards the long-range disorder that is expected for the SmA phase. We now compute CDF with the cylinder length oriented perpendicular to the layer normal (Fig 5D) we consider the local packing of molecules within the layers. Again, the concentric arc-like features–now centred on r = 0 Å–result from the positons of neighbouring molecules; that these do not resolve into point-like features indicates a lack of in-plane positional order.

2,3,6,7,10,11-hexa(hexyloxy)triphenylene (HAT6) is a discotic liquid crystal that exhibits a hexagonal columnar (Col$_h$) mesophase [47, 48]. Unlike the SmA phase exhibited by 8OCB, the Col$_h$ mesophase exhibits significant positional ordering in all three dimensions and so is an excellent test case for CDF analysis. Our simulation of HAT6 is of 500 molecules (72000 atoms) at a temperature of 330 K and a pressure of 1 Bar in a Col$_h$ configuration. Representative instantaneous configurations are shown in Fig 6A and 6B. We computed the CDF for the Col$_h$ phase of our HAT6 simulation with the cylinder length oriented in turn along each Cartesian axes (Fig 6C–6E) using the–vec input argument. We also extend the radius cut-off to 40 Å (using the–R input argument) so as to capture the entirety of the hexagonal lattice. Each CDF plot feature a region of zero probability centred at r = h = 0 Å which results from steric repulsion of molecules.

With the cylinder radius oriented perpendicular to the most prominent feature is found at h = 0 Å, where periodic peaks are found at r with a spacing of ~ 3.5 Å which result from the off-centre and slightly off-parallel nature of the face-to-face stacking of triphenylene. Of interest here, however, are the clear visualisation of multiple neighbouring columns packed in a hexagonal arrangement. It is trivial to obtain the lattice parameter of the Col$_h$ phase from the CDF; we find a value of 18.4 ± 0.5 Å, which compares with an experimental value of 18.7 Å from SAXS experiments [49]. With the cylinder length oriented along the column axis we reveal two dominant features. Firstly, strong on axis peaks at r ≈ 0 Å which are periodic along h, occurring every 3.5 Å, result from face-to-face stacking of the triphenylene core of HAT6. At r ≈ 18 a diffuse feature results from the lateral separation of neighbouring columns.

## Supporting information

**S1 File.**
(DOCX)

## Acknowledgments

Computational work was undertaken on ARC3, part of the high performance computing facilities at the University of Leeds, UK.

## Author Contributions

**Conceptualization:** Richard J. Mandle.

**Data curation:** Richard J. Mandle.

**Formal analysis:** Richard J. Mandle.

**Funding acquisition:** Richard J. Mandle.

**Investigation:** Richard J. Mandle.

**Methodology:** Richard J. Mandle.

**Project administration:** Richard J. Mandle.

**Resources:** Richard J. Mandle.

**Software:** Richard J. Mandle.

**Validation:** Richard J. Mandle.

**Visualization:** Richard J. Mandle.

**Writing – original draft:** Richard J. Mandle.

**Writing – review & editing:** Richard J. Mandle.

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
