## [Decision Letter · Decision Letter 0]

18 Nov 2022

PONE-D-22-23551Implementation of a cylindrical distribution function for the analysis of anisotropic molecular dynamics simulations.PLOS ONE

Dear Dr. Mandle,

Thank you for submitting your manuscript to PLOS ONE. After careful consideration, we feel that it has merit but does not fully meet PLOS ONE’s publication criteria as it currently stands. Therefore, we invite you to submit a revised version of the manuscript that addresses the points raised during the review process. Please submit your revised manuscript by Jan 02 2023 11:59PM. If you will need more time than this to complete your revisions, please reply to this message or contact the journal office at plosone@plos.org. Please include the following items when submitting your revised manuscript:A rebuttal letter that responds to each point raised by the academic editor and reviewer(s). You should upload this letter as a separate file labeled 'Response to Reviewers'.A marked-up copy of your manuscript that highlights changes made to the original version. You should upload this as a separate file labeled 'Revised Manuscript with Track Changes'.An unmarked version of your revised paper without tracked changes. You should upload this as a separate file labeled 'Manuscript'.

We look forward to receiving your revised manuscript.

Kind regards,

Hannes C Schniepp, Dr. sc. nat.

Academic Editor

PLOS ONE

Journal Requirements:

3. Please note that PLOS ONE has specific guidelines on code sharing for submissions in which author-generated code underpins the findings in the manuscript. In these cases, all author-generated code must be made available without restrictions upon publication of the work. Please review our guidelines at https://journals.plos.org/plosone/s/materials-and-software-sharing#loc-sharing-code and ensure that your code is shared in a way that follows best practice and facilitates reproducibility and reuse. New software must comply with the Open Source Definition

“Computational work was undertaken on ARC3, part of the high performance computing facilities at the University of Leeds, UK.  RJM thanks UKRI for funding via a Future Leaders Fellowship (grant no. MR/W006391/1), and the University of Leeds for funding via a University Academic Fellowship. Code for cylindr is available via GitHub (https://github.com/RichardMandle/cylindr)”

Reviewers' comments:

Reviewer's Responses to Questions

**Comments to the Author**

1. Is the manuscript technically sound, and do the data support the conclusions?

Reviewer #1: Partly

Reviewer #2: Partly

2. Has the statistical analysis been performed appropriately and rigorously? 

Reviewer #1: Yes

Reviewer #2: Yes

3. Have the authors made all data underlying the findings in their manuscript fully available?

Reviewer #1: Yes

Reviewer #2: Yes

4. Is the manuscript presented in an intelligible fashion and written in standard English?

Reviewer #1: Yes

Reviewer #2: Yes

5. Review Comments to the Author

Reviewer #1: The manuscript describes analyses of a number of molecular dynamics simulations of anisotropic liquids. The author introduces the cylindrical distribution function (CDF), as a 2-dimensional pendant to the 1-dimensional radial distribution function and argues that more (relevant) information is conveyed in the CDF. The author introduces a python tool to easily perform such analyses, which is made available on github. This is a neat addition, that will be of use to scientists interested in such systems.

I would like to make the following comments:

1. In the introduction, the author describes the radial distribution function as well as parallel and perpendicular components, and orientational variants. This is done in a rather confusing way. I always think of a radial distribution function in terms of the number of particles I find at a particular position (shell of sphere, or a cylinder), relative to the number of particles I would expect in a homogeneous distribution (i.e. the number density times the volume of the shell). This is also the definition the author used for the CDF in equation 6.

a. Below equation 1, the author writes that r is the distance between the selected atoms. This would be r_ij, while r is the coordinate of g(r).

b. Below equation 1, N is described as the number density. As the sum in the equation runs up to N, I suspect N is rather the total number of particles considered. N/V is then the number density, assuming that V is the volume of the box. The author did not introduce the delta function, but let’s assume it is a unitless function that is 1 if its argument is 0 and 0 otherwise.

c. I then, do not see, why there is another division by N, and no correction for the volume of the radial shell (4*pi*r^2*dr). In short, I think the definition of g(r) should read: g(r) = V/(N*4*pi*r^2*dr)*sum_ij(delta(r-r_ij)).

d. Similarly, the description of the symbols under equations 2 and 3 is incomplete: n(r) is not the volume, but the number of particles; what is r_cut, what is r_||?

e. Below equations 4 and 5, the author may want to point out that u_i and u_j are vectors of length 1, and that the equations correspond to the Legendre polynomials.

2. The optimal number of shell increments per Ångstrom (the resolution; page 9) will depend on the number of molecules and the number of snapshots in the trajectories. The more pair-data is available, the finer grained the function can be determined.

3. I cannot follow the description of the very first CDF completely. The authors starts with explaining the high-probability regions at h = +/- 10 Å, r = 0. But the most pronounced maximum at r = 0, is still at h = +/- 5.5 Å, which seems small for the length of the molecule. This particular CDF looks very isotropic to me. Maybe the author can take the readers even more by the hand to emphasize the additional information that can be read from the CDF, compared to the RDF.

4. The bibliography of citation 19 seems incomplete.

Reviewer #2: The Author present a software tool for the evaluation of a bidimensional distribution function in order to study structural properties of fluids with particular emphasis on liquid crystals. The software is made freely available to the community working in the field.

The "cylindrical" distribution function appears to be a useful additional tool to evaluate the structural properties of ordered fluids, so the manuscript is in priciple publishable.

There are, however, a couple of issues that prevents me from recommending publication at this stage.

Considering the first example discussed, the 5CB case, first a general point:

- the Author compares the CDF with the RDF in Figure 2. Obviously the CDF contains a larger amount of information than the RDF, being resolved in two dimensions representing the distance along the director (the height of the cylinder) and the distance perpendicular to the drector (the radius). However does it contain more information than the pair of radial distribution functions resolved along the parallel and perpendicular directions, that is Eq. 2 and Eq. 3?

The second issue is the following: the Author states that "High probability regions at h=±10, r=0 Å result from antiparallel

pairs of molecules, and at h=0, r ~ 5.5 Å from side-by-side pairs." This should be apparent in Figure 2 a) but the bidimensional plot clearly shows that at r=0 the CDF has a maximum at h = +/- 5 (or just larger than 5), not +/- 10 . This seems to be a too short distance for two 5CB molecules on top of each other along the director. This result is not clealy expained.

These comments can be applied to the other examples too, especially the first one, that is: what are the really new additional information dsplayed by the CDF that cannot be found in RDF resolved along an perpendicularly to the director?

6. PLOS authors have the option to publish the peer review history of their article (what does this mean?). If published, this will include your full peer review and any attached files.

Reviewer #1: No

Reviewer #2: No

---

## [Author Response · Author response to Decision Letter 0]

30 Nov 2022

A Formatted response is given in the cover letter; please see below for a copy/paste:

RE:

 “Implementation of a cylindrical distribution function for the analysis of anisotropic molecular dynamics simulations”, submitted for consideration in PLOS One.

Thank you for the review of this manuscript. The comments from both referees are constructive and largely positive, and have strengthened the revised version. I will now address the comments/queries from both referees.

Reviewer #1: The manuscript describes analyses of a number of molecular dynamics simulations of anisotropic liquids. The author introduces the cylindrical distribution function (CDF), as a 2-dimensional pendant to the 1-dimensional radial distribution function and argues that more (relevant) information is conveyed in the CDF. The author introduces a python tool to easily perform such analyses, which is made available on github. This is a neat addition, that will be of use to scientists interested in such systems.

I thank the referee for these supportive comments.

I would like to make the following comments:

1. In the introduction, the author describes the radial distribution function as well as parallel and perpendicular components, and orientational variants. This is done in a rather confusing way. I always think of a radial distribution function in terms of the number of particles I find at a particular position (shell of sphere, or a cylinder), relative to the number of particles I would expect in a homogeneous distribution (i.e. the number density times the volume of the shell). This is also the definition the author used for the CDF in equation 6.

a. Below equation 1, the author writes that r is the distance between the selected atoms. This would be r_ij, while r is the coordinate of g(r).

b. Below equation 1, N is described as the number density. As the sum in the equation runs up to N, I suspect N is rather the total number of particles considered. N/V is then the number density, assuming that V is the volume of the box. The author did not introduce the delta function, but let’s assume it is a unitless function that is 1 if its argument is 0 and 0 otherwise.

c. I then, do not see, why there is another division by N, and no correction for the volume of the radial shell (4*pi*r^2*dr). In short, I think the definition of g(r) should read: g(r) = V/(N*4*pi*r^2*dr)*sum_ij(delta(r-r_ij)).

d. Similarly, the description of the symbols under equations 2 and 3 is incomplete: n(r) is not the volume, but the number of particles; what is r_cut, what is r_||?

e. Below equations 4 and 5, the author may want to point out that u_i and u_j are vectors of length 1, and that the equations correspond to the Legendre polynomials.

Firstly I want to apologise as the introduction to the manuscript is – on reflection – not well written, with key information missing, incomplete description of equations, and errors. I can only apologise for presenting such sloppy work. I will address each comment – (a) through to (e) in turn.

 The referee is quite correct here that in equation 1 N is the total number of particles considered and not the number density as is written in the text. The erroneous text has been changed:

 “…N is the number of molecules in the simulation…”

 And additionally, the delta in eq. 1 has been defined in the text as the Dirac delta function:

 “…and δ is the Dirac delta function.”

 The referee is correct, eq. 1 should read:

g(r)=V/(N4πr^2 ∆r) ⟨├ ∑_ij^N▒δ(r-r_ij ) ⟩┤ eq. 1

And I also include text below:

“…, r is the radius of the spherical shell, ∆r is the thickness of the spherical shell, …”

 The referee is also correct that, in reference to equations 2 and 3, n(r) is described in the text as being “the volume of the molecules contained within a cylindrical volume…” which is not correct, rather n(r) is the “number of molecules contained within a cylindrical volume…”, and the manuscript text has been amended to reflect this.

r_cutis defined in the text as:

“…where r_cut is the cut-off value used for calculating the distribution function.”

 Text has been added (additions highlighted in yellow):

“Where u_i and u_j are two vectors of unit length describing the orientation of the ith and jth molecules, respectively, and r is the distance between the centre-of-mass (or selected atoms) of these same two molecules, with eq. 4 and eq. 5 corresponding to the Legendre polynomials.”

2. The optimal number of shell increments per Ångstrom (the resolution; page 9) will depend on the number of molecules and the number of snapshots in the trajectories. The more pair-data is available, the finer grained the function can be determined.

Yes this is correct and an interesting point, in future we could set a ‘default’ resolution on the fly depending on how fine the available trajectory data is, but I consider this something to add to a future version of the code. Of course we need to provide a means for a user specified value as calculating the CDF for very large simulations the Numpy arrays can consume a lot of RAM, and decreasing the resolution is one way to get around this.

3. I cannot follow the description of the very first CDF completely. The authors starts with explaining the high-probability regions at h = +/- 10 Å, r = 0. But the most pronounced maximum at r = 0, is still at h = +/- 5.5 Å, which seems small for the length of the molecule. This particular CDF looks very isotropic to me. Maybe the author can take the readers even more by the hand to emphasize the additional information that can be read from the CDF, compared to the RDF.

On re-reading the manuscript I share this view; after recalculating the CDF taking various cuts, I find I had originally included a CDF with the cylinder length perpendicular to the director rather than parallel to it. I have recomputed the CDF – both parallel and perpendicular – for the same trajectory, and included an additional graphic that shows the difference (Fig 1b). 

The text has been amended (additions in yellow):

 “Both CDF orientations show a region of zero probability centred at h=r=0 Å, which results from steric repulsion of molecules. High probability regions at h=±13, r=0 Å and h=±18, r=0 Å result from antiparallel pairs of molecules, and at h=0, r ~ 5.5 Å from side-by-side pairs. A series of arcs centred on h=0 Å occur every r ~5.5 Å as a consequence of side-by-side packing of molecules; the first arc being the neighbouring molecule(s), the second being the neighbour of neighbours and so on. The diffusivity of all features indicates a lack of positional order, as expected for the nematic state. For the CDF oriented perpendicular to the director we see diffuse concentric rings every r ~5.5 Å , the origins of these being neighbouring molecules as discussed above.”

Following the comments of Referee 2, I have also calculated the RDF for this trajectory of 5CB and resolved into its parallel and perpendicular components. Even for a this simple case (a conventional nematic) we can see that the CDF provides extra information, with the overlap of on-axis and off-axis features which result from quite subtle close-contacts between neighbours-of-neighbours.

New text, updated figure 2 and caption below:

“By resolving the RDF into components perpendicular and parallel to the director (Fig. 2f) we can see the same features present in the CDF. However, resolving the RDF in this way neglects off-axis interactions; a rather subtle example is the overlap of the ‘on-axis’ arcs (at h~ 10-12, r ~ 0-6 Å) with the off-axis arc that results from the lateral ‘neighbour-of-neighbours’ (at h ~ -10-10, r ~ 6-9 Å). This subtle anisotropy in local positioning manifests in variations in intensity in the concentric rings in the perpendicular CDF plot (Fig. 2e) which is not apparent from the resolved RDF (Fig. 2f).

Figure 2: (a) Molecular structure and transition temperatures (°C) of 5CB; (b) instantaneous configuration of 5CB in the nematic phase at 293 K, the simulation has <P2> = 0.56 ± 0.019 over the 50 ns MD run; (c) schematic showing the nematic director from (b) as a red arrow, and (exaggerated) cylindrical shells whose radius (∆r) forms a plane orthogonal (∆r_(||)) or parallel (〖∆r〗_⊥) to the director, used to calculate the CDF parallel and perpendicular to the director, respectively; (d) parallel and (e) perpendicular CDF plots for the nematic simulation of 5CB as generated by cylindr; (f) radial distribution functions (RDF) for the same simulation, with the RDF resolved into parallel and perpendicular components according to eq. 2 and eq. 3.

“

4. The bibliography of citation 19 seems incomplete.

Reference 19 has been corrected.

Reviewer #2: The Author present a software tool for the evaluation of a bidimensional distribution function in order to study structural properties of fluids with particular emphasis on liquid crystals. The software is made freely available to the community working in the field.

The "cylindrical" distribution function appears to be a useful additional tool to evaluate the structural properties of ordered fluids, so the manuscript is in priciple publishable.

There are, however, a couple of issues that prevents me from recommending publication at this stage.

Considering the first example discussed, the 5CB case, first a general point:

- the Author compares the CDF with the RDF in Figure 2. Obviously the CDF contains a larger amount of information than the RDF, being resolved in two dimensions representing the distance along the director (the height of the cylinder) and the distance perpendicular to the drector (the radius). However does it contain more information than the pair of radial distribution functions resolved along the parallel and perpendicular directions, that is Eq. 2 and Eq. 3?

The referee raises an important point. I would suggest that the CDF does yield additional information than the RDF resolved into two dimensions (as in Eq. 2, Eq. 3) as the explicit spatial relation between features is preserved. When resolving the RDF in this way we lose positional information in the ‘other’ direction, e.g. if we resolve the RDF parallel to the director then we neglect information about radial separation (and vice versa). 

I have added text on pg. 2:

 “… and so information about the spatial relationship between features is preserved.”

The response to the next comment (below) also addresses some of these highlighted issues.

The second issue is the following: the Author states that "High probability regions at h=±10, r=0 Å result from antiparallel

pairs of molecules, and at h=0, r ~ 5.5 Å from side-by-side pairs." This should be apparent in Figure 2 a) but the bidimensional plot clearly shows that at r=0 the CDF has a maximum at h = +/- 5 (or just larger than 5), not +/- 10 . This seems to be a too short distance for two 5CB molecules on top of each other along the director. This result is not clealy expained.

This was the result of my including the wrong CDF image (discussed in response to reviewer #1 also); in the original submission the presented CDF was for a cylinder oriented perpendicular to the director, not parallel as should have been the case. This has been corrected, and the revised plot shows the expected behaviour. The new Figure and caption reads:

Figure 2: (a) Molecular structure and transition temperatures (°C) of 5CB; (b) instantaneous configuration of 5CB in the nematic phase at 293 K, the simulation has <P2> = 0.56 ± 0.019 over the 50 ns MD run; (c) schematic showing the nematic director from (b) as a red arrow, and (exaggerated) cylindrical shells whose radius (∆r) forms a plane orthogonal (∆r_(||)) or parallel (〖∆r〗_⊥) to the director, used to calculate the CDF parallel and perpendicular to the director, respectively; (d) parallel and (e) perpendicular CDF plots for the nematic simulation of 5CB as generated by cylindr; (f) radial distribution functions (RDF) for the same simulation, with the RDF resolved into parallel and perpendicular components according to eq. 2 and eq. 3.

These comments can be applied to the other examples too, especially the first one, that is: what are the really new additional information dsplayed by the CDF that cannot be found in RDF resolved along an perpendicularly to the director?

I agree that this could have been illustrated better; clearly there are some instances where the CDF is superior to such a ‘resolved’ RDF, and some where is merely equal to them. We use 5CB as an exemplar to show how the CDF can capture additional detail to the resolved RDF case:

“By resolving the RDF into components perpendicular and parallel to the director (Fig. 2f) we can see the same features present in the CDF. However, resolving the RDF in this way neglects off-axis interactions; a rather subtle example is the overlap of the ‘on-axis’ arcs (at h~ 10-12, r ~ 0-6 Å) with the off-axis arc that results from the lateral ‘neighbour-of-neighbours’ (at h ~ -10-10, r ~ 6-9 Å). This subtle anisotropy in local positioning manifests in variations in intensity in the concentric rings in the perpendicular CDF plot (Fig. 2e) which is not apparent from the resolved RDF (Fig. 2f).

Of course, there will be cases which we have not thought of in which the CDF is superior to the resolved RDF, and this is our motivation for describing this software tool and releasing it for use by the community. 

Finally, the editorial office asked for clarification regarding the funding of this work. 

In response:

 UK Research and Innovation (UKRI) Future Leaders Fellowship to R. Mandle (grant no. MR/W006391/1); University of Leeds, University Academic Fellowship to R. Mandle (no grant number).

 I receive my salary from the University of Leeds.

 We did receive funding for this work, see (a).

Yours Sincerely

Dr. Richard J. Mandle

UKRI Future Leaders Fellow & 

University Academic Fellow 

School of Chemistry & School of Physics and Astronomy 

University of Leeds, Leeds, UK, LS2 9JT

---

## [Decision Letter · Decision Letter 1]

12 Dec 2022

Implementation of a cylindrical distribution function for the analysis of anisotropic molecular dynamics simulations.

PONE-D-22-23551R1

Dear Dr. Mandle,

We’re pleased to inform you that your manuscript has been judged scientifically suitable for publication and will be formally accepted for publication once it meets all outstanding technical requirements.

Kind regards,

Hannes C Schniepp, Dr. sc. nat.

Academic Editor

PLOS ONE

Additional Editor Comments (optional):

Reviewers' comments:

Reviewer's Responses to Questions

**Comments to the Author**

1. If the authors have adequately addressed your comments raised in a previous round of review and you feel that this manuscript is now acceptable for publication, you may indicate that here to bypass the “Comments to the Author” section, enter your conflict of interest statement in the “Confidential to Editor” section, and submit your "Accept" recommendation.

Reviewer #1: All comments have been addressed

Reviewer #2: All comments have been addressed

2. Is the manuscript technically sound, and do the data support the conclusions?

Reviewer #1: Yes

Reviewer #2: Yes

3. Has the statistical analysis been performed appropriately and rigorously? 

Reviewer #1: N/A

Reviewer #2: Yes

4. Have the authors made all data underlying the findings in their manuscript fully available?

Reviewer #1: Yes

Reviewer #2: Yes

5. Is the manuscript presented in an intelligible fashion and written in standard English?

Reviewer #1: Yes

Reviewer #2: Yes

6. Review Comments to the Author

Reviewer #1: (No Response)

Reviewer #2: (No Response)

7. PLOS authors have the option to publish the peer review history of their article (what does this mean?). If published, this will include your full peer review and any attached files.

Reviewer #1: No

Reviewer #2: No

---

## [Editor Report · Acceptance letter]

19 Dec 2022

PONE-D-22-23551R1 

Implementation of a cylindrical distribution function for the analysis of anisotropic molecular dynamics simulations. 

Dear Dr. Mandle:

I'm pleased to inform you that your manuscript has been deemed suitable for publication in PLOS ONE. Congratulations! Your manuscript is now with our production department. 

Kind regards, 

on behalf of

Dr. Hannes C Schniepp 

Academic Editor

PLOS ONE